# Passenger Routing Algorithm for COVID-19 Spread Prevention by Minimising Overcrowding

Dimitrios Tolikas, Evangelos D. Spyrou * and Vassilios Kappatos

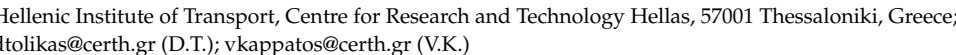

Hellenic Institute of Transport, Centre for Research and Technology Hellas, 57001 Thessaloniki, Greece; dtolikas@certh.gr (D.T.); vkappatos@certh.gr (V.K.)
* Correspondence: espyrou@certh.gr

**Abstract:** COVID-19 has become a pandemic which has resulted in measures being taken for the health and safety of people. The spreading of this disease is particularly evident in indoor spaces, which tend to get overcrowded with people. One such place is the airport where a plethora of passengers gather in common places, such as coffee shops and duty-free shops as well as toilets and gates. Guiding the passengers to less overcrowded places within the airport may be a solution to reduce disease spread. In this paper, we suggest a passenger routing algorithm whereby the passengers are guided to less crowded places by using a weighting factor, which is minimised to accomplish the desired goal. We modeled a number of shops in an airport using the AnyLogic software and we tested the algorithm showing that the exposure time is less with routing and that people are appropriately spread out across the common spaces, thus preventing overcrowding. Finally, we added a real airport in Kavala, Greece to show the efficiency of our approach.

**Keywords:** routing; weight; COVID-19; overcrowding; virus; passenger

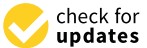


## 1. Introduction

COVID-19 and other airborne-transmitted diseases have significant impacts on human health, even leading to death. COVID-19 is considered a highly transmissible disease [1]. Particularly in enclosed spaces, areas with crowding often increase the risk of epidemic spread of infections. Airports are crowded places where people move around duty-free shops, restaurants, and gates. Therefore, there is a need to organize people's movement within the airport to avoid crowding. Two recent studies demonstrated modeling the spread and human behavior for the transmission of COVID-19 in indoor spaces, respectively [2,3]. The virus transmission occurs either through droplets or via direct contact of an individual with another person [4].

To prevent the transmission of COVID-19 through the air, it is necessary to maintain a specific number of individuals per square meter. Consequently, it is important to organize the passenger flow heading towards or already present in various areas of the airport in order to minimise the risk of virus transmission. Passenger counting can be conducted from the moment they disembark the plane or perform check-in and can continue throughout their journey to different airport areas.

The investigation of passenger flow in airport terminals involves various methods. The conventional method primarily relies on on-site surveys, including passenger counting and questionnaires, which are used to validate model results or signals from sensors [5]. However, this approach demands extensive work, particularly in large airports. To address this, indoor sensors and technologies like Wi-Fi information [6], mobile phone data [7], RFID [8], motion sensors [9], PIR sensors [10], and surveillance videos [11] have been employed in airports and other buildings. Simulation models have also been developed to analyze and predict passenger flow in airport terminals.

The flow of passengers in an airport terminal comprises two main components: service counters where passengers stay and movements between these counters. Queuing theory

is used to describe how passengers wait in line at service counters like at check-in, security, and boarding [12]. The transfer conditional probability table is a common mathematical method to analyze interactions and transfer probabilities between each counter [13,14].

Routing of passengers can be of extreme importance in indoor spaces in order to avoid overcrowding. As such, an efficient scheme is mandatory to guide passengers in airports to particular common spaces in a manner that promotes sparsity. Since passengers flow through the airport in large numbers, attempting to keep them safe and distanced in common spaces is of the utmost importance.

This paper explores the domain of simulation using the sophisticated features offered by the AnyLogic software. Our simulation framework is designed to capture the intricacies inherent in both indoor complexes and the dynamic environments found in airports. Central to our methodology is the creation and application of a straightforward yet highly efficient weighting factor, which, when deployed, acts as a pivotal catalyst, yielding effective outcomes in the domain of passenger routing.

A foundational assumption guiding our research involves the acquisition of data through a dual-channel approach, specifically employing surveillance cameras and extracting nuanced preferences from mobile phones. Through the synergistic use of these data sources, our aim is to construct a comprehensive representation of passenger behaviors and interactions within the simulated spaces.

At the core of our strategic approach is the judicious guidance of individuals toward common areas within the airport. Leveraging the capabilities of the weighting factor, we endeavor to optimize the movement of passengers in a manner that mitigates congestion and, consequently, reduces the potential exposure time to viruses.

This paper is structured as follows: Section 2 provides the related work, Section 3 gives the COVID-19 transmission simulation modeling, Section 4 provides the simulation modeling of transmission inside stores with passenger routing and without, Section 5 provides the simulation modeling of a small airport, Section 6 provides the simulation modeling of transmission of COVID-19 at LGKM Kavala Airport, and Section 7 provides the conclusions.

## 2. Related Work

Due to the complexity and size of airport terminals, agent-based simulation (ABS) models are employed to understand continuous changes in passenger flow and distribution. ABS treats each passenger as an individual research subject, using a social force model to control their movements [15].

In [13], simulation has proven instrumental in understanding and assessing passenger movement during airport departure procedures. The authors' methodology not only facilitates the evaluation but also the prediction of airport operational efficiency. Its application supports airport management in identifying operational bottlenecks, specifically related to challenges in flight schedule planning. Furthermore, it provides precise insights into how changes in infrastructure and operations impact airport functionality. In that study, they used simulation to examine various load factors associated with diverse flight schedules. The outcomes of the simulation emphasize the significant influence of the flight schedule on passenger flows. The proposed simulation framework and model show promise in foreseeing the effects of different flight schedules, serving as a proactive tool to refine them before implementation. These findings suggest that integrating the creation of flight schedules with passenger simulation analysis could effectively address challenges in managing passenger flow within airport terminals.

In [16], the AnyLogic software serves to simulate human behavior within specific building structures, generating valuable data on people flow. These data are pivotal in establishing a people flow model, subsequently applied to estimate human occupancy through the utilization of a Kalman filter. An initial simulation involving a single room and corridor demonstrates the superior performance of the Kalman filter estimation, based on the identified model, compared to estimations relying solely on sensors. Furthermore, this

methodology is substantiated through a real-world experiment where authentic cameras and beam sensors are installed in a corridor and room. The results of this practical experiment reinforce the efficacy of the estimation technique which combines the Kalman filter and AnyLogic, surpassing the performance of exclusive reliance on sensors. This proposed method offers a viable solution to the challenge of model identification for estimating building occupancy, particularly in scenarios where real data on people flow are limited.

In [17], the authors utilized AnyLogic to analyze passenger flow at the entrance of Wulin Station. By comparing various quantities of ticket windows based on different pedestrian arrival rates, they reached a conclusion: during high-traffic hours (with a pedestrian arrival rate of 2500/h), it is more efficient to open four ticket windows. Conversely, during low-traffic hours (with a pedestrian arrival rate of 1500/h), it is preferable to operate two ticket windows. It is important to note that due to time constraints and the closure of other subway lines in Hangzhou, the precise statistics regarding pedestrian arrival rates during peak and off-peak hours were not determined in this article. The simulation model employed in that study can not only be applied to other subway entrances but is also easily adaptable for altering the pedestrian arrival rate.

In [18], the paper introduces a methodology aimed at modeling the movement of entities within a hub airport, demonstrated through the simulation of the New Barcelona International Airport, a comprehensive and intricate case study. The principles delineated in this approach are transferable to other airports with similar configurations. The simulation of entities' movement within a hub airport heavily relies on accurately interpreting diverse data types. The article meticulously delineates the categorizations of these components, the primary model parameters, and their role in establishing a presentation format for incoming entities. To simulate the movement of these entities within the airport, the proposal involves employing a Simple Reflexive agent, providing a detailed analysis of the time and delays arising from their actions. The methodological approach leans on the Specification and Description Language (SDL), a widely acknowledged formal graphical and standard language. In the highlighted case study, SDL played a crucial role, acting as a primary facilitator for effective communication among all stakeholders.

In [19], researchers introduced a simulation approach employing AnyLogic software to simulate the movement patterns of passengers both entering and exiting a metro station, with a specific emphasis on a specific line. The simulation logic was divided into three core components: the inflow of passengers, outflow of passengers, and the arrival of trains. Train arrivals were synchronized with the generation of passengers using hourly flow distributions. Different scenarios of passenger flow distributions were tested to determine the optimal number of functioning ticket windows. Selection criteria focused on ensuring that ticket level waiting times aligned with train intervals and that ticket offices utilized their staff efficiently. The simulation employed both 2D and 3D perspectives to visualize pedestrian behavior and identify congested areas. Analyzing these flow patterns led to suggestions for optimizing exit and entrance gates. The model effectively demonstrated its ability to evaluate the overall operational status while also proposing practical improvements for the organization and layout of the facility.

In [20], the authors address the effective management of an Emergency Department (ED) which involves navigating a highly intricate landscape, given the admission of patients with a spectrum of ailments and varying degrees of urgency. This complexity demands the coordination of diverse activities, encompassing both human and medical resources. The nuanced nature of ED management poses challenges, notably overcrowding, which can adversely impact the quality and accessibility of healthcare services.

The aforementioned study strategically employs Process Mining techniques within a tangible case study, focusing on the operations of the ED. Leveraging the ED database, advanced discovery techniques are applied to unravel potential patient pathways based on information acquired during the triage process. The overarching goal is to generate precise process models that not only facilitate the replication of ED workflows but also enable the prediction of patient trajectories within this dynamic healthcare setting.

### 3. COVID-19 Transmission Simulation Modeling

According to Chen et al. [4], COVID-19 can spread in two primary ways (Figure 1). The first method of transmission involves droplets, and the second involves direct contact between a virus carrier and a healthy individual. There are two subcategories of droplet transmission, depending on their sizes. The first subcategory includes close contact between a carrier and a healthy individual as well as contact with a contaminated surface when the droplets are larger than 5 μm. When the droplets are smaller than 5 μm, COVID-19 is transmitted via the aerosol route.

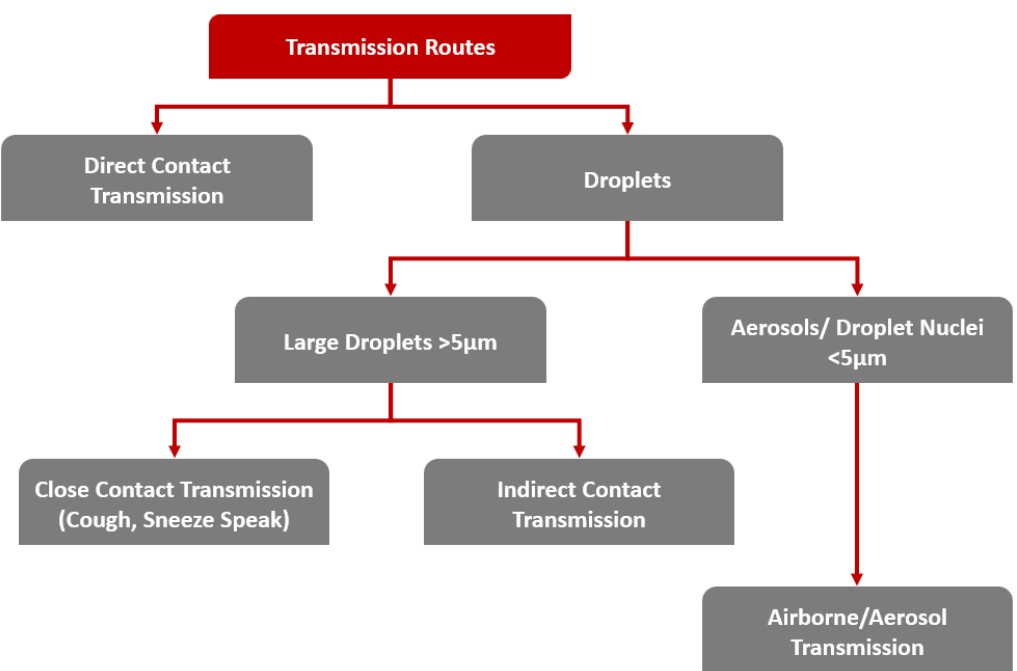

**Figure 1.** COVID-19 regarding transmission routes dropdown (red colour).

In the model presented in this research, the transmission of COVID-19 between passengers occurs only through close contact. Specifically, in the model it is assumed that a passenger who is a carrier of the virus can transmit the virus either if he comes too close to a healthy passenger or if he coughs or sneezes at a certain distance from the healthy passenger. According to the recommendations of the World Health Organization, anyone who comes into contact with a possible or confirmed carrier of COVID-19 at a distance of less than or equal to 1 m for 15 min is also considered to be a possible case [21]. Therefore, in the model, we consider that an individual who has COVID-19 can transmit the virus over a distance of less than or equal to 1 m. In the case of transmission of the virus through coughing or sneezing, we consider a healthy passenger to have become infected when a passenger carrying the virus coughs or sneezes within a radius of less than or equal to 2.5 m [22]. Also, we consider the contagiousness of a sneeze/cough to last 15 s, and a passenger carrying the virus is considered to cough or sneeze every 15–20 s. During the simulation, the cumulative exposure time is used to calculate the exposure of a healthy passenger to the virus according to Formula (1):

$$T_{exposure} = T_{closeContact(\leq 1m)} + T_{cough/sneeze} \text{ [s],} \tag{1}$$

where $T_{closeContact(\leq 1m)}$ is the total exposure time of close contact ($\leq 1$ m) of a healthy passenger with a passenger carrying the virus, and

$T_{cough/sneeze}$ is the total exposure time of a healthy passenger to a cough/sneeze from a carrier of the virus.

It should be highlighted that in the model there are no asymptomatic carriers of the virus and furthermore all carriers can transmit the virus with the same probability.

### 4. Simulation Modeling of Transmission inside Stores with Passenger Routing and without

Figure 2 shows the floor plan of an airport area with stores, in which the transmission of COVID-19 will be modeled among passengers. As can be seen in Figure 2, there are four types of stores and two stores per type. The stores may represent restaurants, coffee shops, duty-free shops, bathrooms, and any other store that we can find in a typical airport.

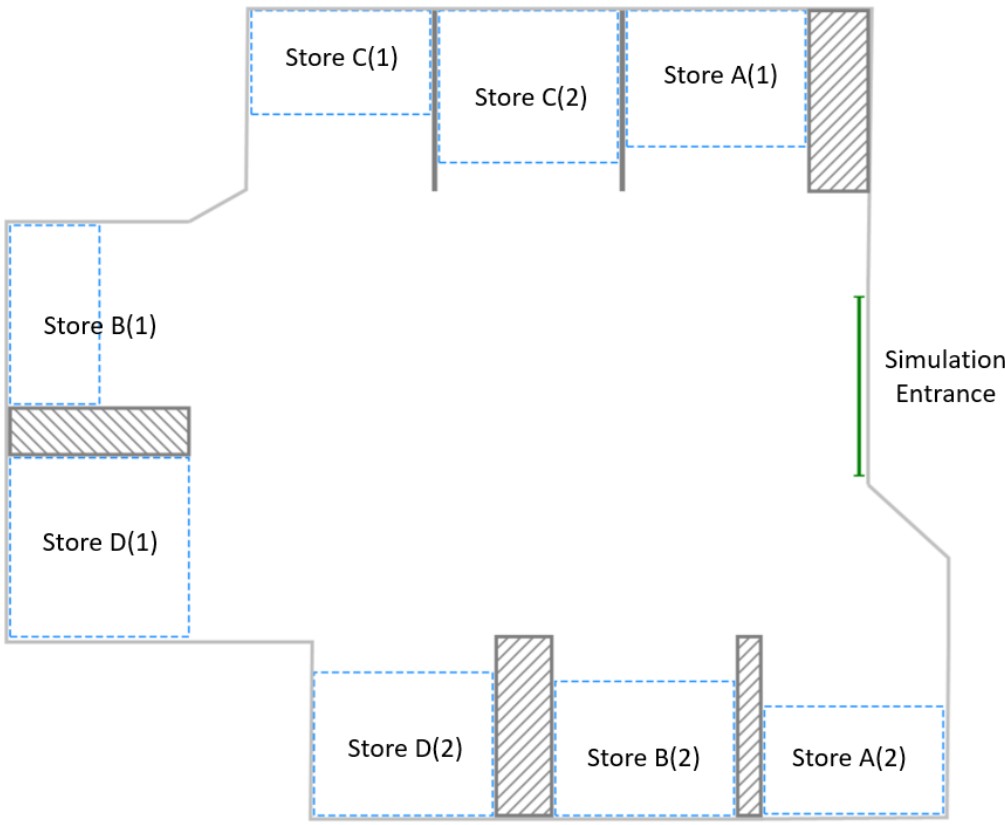

**Figure 2.** Stores of a simulated airport floor plan.

Each passenger, after entering the airport area with the stores, has the option to visit each of the four types of store based on their preferences. It should be noted that each passenger can visit only one store of each specific type and only one time. Once the passenger has visited all the stores they wanted to visit, the passenger exits this area of the airport.

Each store has a maximum occupancy limit depending on the square meters of the store. If the store reaches its maximum capacity, then the passenger who wants to enter waits outside the store until a passenger inside the store leaves. The maximum occupancy limit of the stores is set by dividing the square meters of the store by 4 (4 square meters for every passenger). During COVID-19, people in the majority of countries were required to adhere to the 4-square-meter rule in indoor spaces.

#### 4.1. Simulation of COVID-19 Transmission between Passengers in an Airport Area with Stores

For the simulation, the user must provide some input data to the model. In particular, the user must specify how many passengers will enter the airport area with the stores, the time delay between each passenger's arrival, and how much time they will spend in each type of store. The input data are displayed in Table 1.

It should be noted that the time spent by passengers in each of the stores is not based on real data. However, in the context of the comparison presented in this research, the result is not affected by these values because they are fixed at every simulation.

**Table 1.** Store simulation model input data.

| Input Data | Value |
|:---:|:---:|
| Number of passengers | 1000 |
| Inter-arrival time | 1 passenger per minute |
| Time spent at Clothing Stores | 10 min |
| Time spent at Coffee Shops | 5 min |
| Time spent at Restaurants | 15 min |
| Time spent at Bathrooms | 3 min |

For the simulation, a list of passengers must be given as input from the user to the model, which will include the preferences (Store A, Store B, Store C, Store D) of each passenger and whether they are carriers of COVID-19. In the context of this study, a comparison of the transmission of COVID-19 amongst passengers is made, in the case of passengers visiting the stores in a random order versus visiting the stores in a preset order calculated by a routing algorithm. For that reason, a Python script was developed in order to generate a list of passengers with random preferences and a random order of store visits.

In the case of passenger routing in an airport area with stores, an algorithm was developed that directs passengers based on their preferences but also based on the number of passengers in each store at a specific time, the number of passengers waiting outside the stores, and the number of passengers en route to each store.

This algorithm is inspired by the theory of stochastic network optimization and specifically the work described in [23] regarding queues. It is essentially an algorithm inspired by the maxWeight algorithm [24] that resembles the backpressure algorithm of [25]. Stochastic queue optimization involves applying random or probabilistic methods to optimize queues or waiting lines, common in systems like computer networks, transportation, and service industries. Unlike traditional queuing theory, which assumes fixed arrival and service rates, stochastic queue optimization considers the inherent uncertainty and randomness in these rates due to real-world fluctuations. The goal is to improve the efficiency and resource utilization of queuing systems by using probabilistic models and optimization techniques. This approach addresses variability in demand, random service times, and unpredictable disruptions, aiming to minimise waiting times, reduce congestion, and allocate resources effectively, especially in dynamic and unpredictable environments where deterministic models may be insufficient.

The algorithm also takes into account the maximum allowed capacity of each store (or bathroom) as well as the time that one passenger spends at a store. When a passenger enters the store area of the airport, the algorithm calculates a weighting factor for each store and then, based on the passenger's preferences, ranks the stores in ascending order based on this factor. After calculating the weighting factors, the algorithm directs the passenger to the option with the lowest weighting factor.

When the passenger exits the store, the algorithm recalculates the weighting factors for each store that the passenger wants to visit next, and if a better sequence of store visits is found, then the routing is updated and the passenger goes next to the option with the lowest coefficient risk. After the passengers finish visiting all the stores they want to visit, they exit the area. The formula for calculating the weighting factor for each store is given in (2) [26].

$$F_i = \begin{cases} (P_{i,w} + P_{i,g} + \frac{P_{i,in}}{P_{i,max}}) \times T_i, & P_{i,max} = P_{i,in} \\ (P_{i,g} + \frac{P_{i,in}}{P_{i,max}}) \times T_i - \alpha, & P_{i,max} > P_{i,in} \end{cases} \tag{2}$$

where

$P_{i,w}$ is the number of passengers waiting to enter the store $i$;

$P_{i,g}$ is the number of passengers en route to the store $i$;

$P_{i,in}$ is the number of passengers inside the store $i$;

$P_{i,max}$ is the maximum allowed capacity of the store $i$;

$T_i$ is the time that one passenger spends in the store $i$;

$\alpha$ is a negative constant (e.g., $-10{,}000$.)

The constant $\alpha$ is included in the formula in order to route the passenger to stores that are not full (maximum capacity limit).

### 4.2. Results

For the simulation, a sample of passengers was given as input to the model from which:

- 85% of passengers will want to shop for clothing;
- 65% of passengers will want to eat;
- 70% of passengers will want to drink coffee;
- 75% of passengers will want to go to the bathroom.

Regarding the percentage of passengers who have COVID-19, simulations were performed with percentages of passenger infection rates of 2%, 4%, 7%, 10%, 15%, and 20%. Also, for each percentage of infected passengers, 40 simulations (20 simulations without passenger routing and 20 simulations with passenger routing) were performed and the results presented below are the averages of the results.

The results of the simulations are given below in Figure 3. In particular, Figure 3 shows the total exposure time of healthy passengers to COVID-19 inside the airport area with the stores for the case of passengers randomly visiting the stores and for the case of passengers routed to shops for the different passenger infection rates. According to the results, it is observed that by using the algorithm to guide the passengers in every case, there is a reduction in the transmission of the virus.

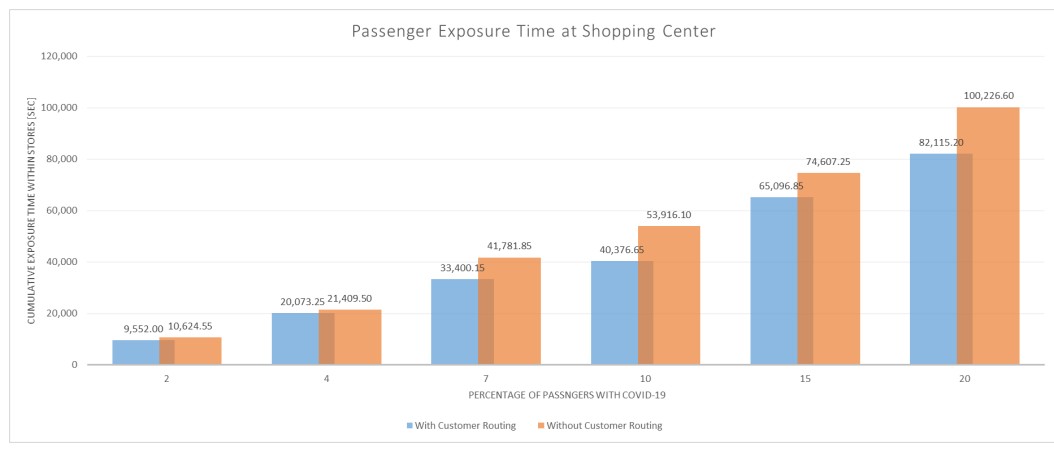

**Figure 3.** Passenger exposure time in airport area with stores.

The maximum number of passengers per shop and simulation with and without routing are shown in Table 2. According to Figure 4, with the routing of passengers in the considered airport area, a decrease in the maximum number of passengers in each store can be seen. For instance, in Store A (Store A (1) and Store A (2)), for an infected passenger rate of 2%, the average maximum number of passengers in the case without passenger guidance is 12.85 and 11.35, while in the case with passenger routing it is 7.35 and 8.30, respectively. The results show that passenger navigation inside the airport area with stores contributes considerably to preventing congestion in the stores and, as a result, lessens the spread of COVID-19.

**Table 2.** Airport simulation model input data.

| Simulation Variables | Value |
|---|---|
| Start Boarding Time | 40 min before Take-Off |
| Flight Check-In Start Time | 150 min before Take-Off |
| Number of Passengers per Flight | 120 |
| Total Number of Passengers | 1560 |
| Arrival Rate | 100 Passengers per Hour |
| Time spent at Other shops | 10 min |
| Time spent at Coffee Shops | 5 min |
| Time spent at Restaurants | 15 min |
| Time spent at Bathroom | 3 min |

| Infection Rate (%) | Without Passenger Routing | | | | | | | | With Passenger Routing | | | | | | | |
|---|---|---|---|---|---|---|---|---|---|---|---|---|---|---|---|---|
| | Store A (1) | Store A (2) | Store B (1) | Store B (2) | Store C (1) | Store C (2) | Store D (1) | Store D (2) | Store A (1) | Store A (2) | Store B (1) | Store B (2) | Store C (1) | Store C (2) | Store D (1) | Store D (2) |
| 2 | 12.85 | 11.35 | 12.20 | 15.45 | 7.75 | 6.65 | 5.40 | 6.25 | 7.35 | 8.30 | 9.40 | 8.10 | 4.05 | 5.25 | 3.40 | 3.10 |
| 4 | 11.10 | 11.15 | 11.70 | 10.65 | 7.00 | 6.15 | 5.00 | 5.55 | 7.15 | 8.20 | 9.25 | 7.90 | 3.90 | 5.10 | 3.15 | 3.05 |
| 7 | 11.00 | 10.65 | 12.45 | 12.95 | 6.75 | 8.30 | 6.65 | 5.25 | 7.30 | 8.45 | 8.90 | 7.75 | 3.95 | 5.00 | 3.35 | 3.25 |
| 10 | 12.45 | 11.10 | 10.40 | 12.00 | 7.05 | 7.00 | 5.80 | 4.95 | 7.20 | 8.40 | 9.00 | 7.90 | 4.00 | 5.05 | 3.20 | 3.10 |
| 15 | 9.75 | 11.65 | 11.05 | 10.90 | 7.40 | 6.15 | 6.95 | 0.00 | 7.60 | 8.40 | 9.40 | 7.85 | 3.90 | 5.15 | 3.40 | 2.90 |
| 20 | 10.90 | 11.00 | 11.20 | 12.00 | 6.65 | 6.70 | 6.40 | 6.50 | 7.40 | 8.50 | 9.65 | 8.25 | 4.05 | 5.15 | 3.40 | 3.15 |

**Figure 4.** Maximum number of passengers per store with and without routing.

## 5. Simulation Modeling of Small Airport

In this section, the transmission of COVID-19 between passengers in a virtual airport is simulated. The logic followed is the same as that of the transmission of COVID-19 between passengers in an airport area with stores but in this simulation model we model the whole airport. Figure 5 shows the floor plan of the airport. The airport stores consist of two restaurants, two coffee shops, two bathrooms, and two other shops (e.g., duty-free shops, money exchange shops, etc.)

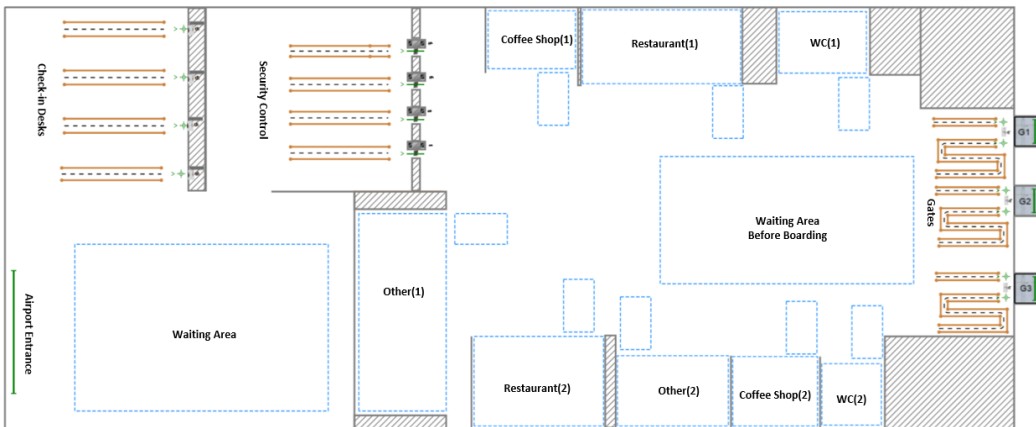

**Figure 5.** Airport stores floor plan.

The airport passengers enter the airport through the entrance. The logic diagram that each passenger follows is given in Figure 6. Each passenger in the model has the option to check in online before arriving at the airport or to check in at the check-in desks inside the airport. We assume that there is no passenger who has checked in online and wishes to hand over his/her suitcase upon arrival at the airport.

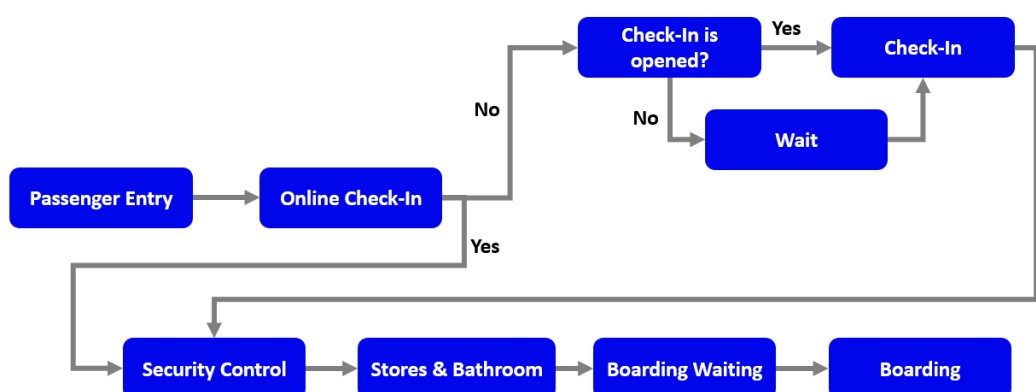

**Figure 6.** Passenger logic diagram.

Therefore, upon entry, the model checks whether the passenger has checked in online or not. If the passenger has checked in online, the passenger proceeds to the security control area; otherwise, the model checks whether the check-in desk has opened for the passenger's flight number. If the check-in desk has not opened, the passenger will have to wait in the waiting area until it opens. In the case where the check-in desk is open, the passenger can proceed to the check-in desks. The passenger then proceeds to the security control area. After check-in, the passengers can visit one of the shops according to their preferences, use the bathroom, or wait in the waiting area until boarding of his/her flight begins.

It is important to emphasize that in the event that the passenger has passed the control and his/her flight begins boarding, the passenger goes to the boarding area as soon as the activity he/she is doing at that time is finished. Also, in the model, passengers are separated into those travelling with economy class tickets and those travelling with business class tickets. The main difference is that during check-in and during boarding, they are served in different queues.

For the simulation, the user must provide some input data to the model. In particular, the user states the boarding start time before the departure of each flight, the check-in start time of each flight, the number of passengers on each flight, the total number of simulated passengers, and also the passenger visit times inside the stores. In Table 2 below, the input

variables of the model are given as well as their corresponding values which were used in the context of this research.

In addition to the variables given in Table 2, the flights' departure schedule is also given as input to the model. Specifically, for each flight, the name, the departure time, and the departure gate are given. As shown in Table 3, the user enters the number of passengers of each flight. When a passenger enters the model, the passenger takes a seat from the next available flight. When a flight reaches its maximum capacity (depending on the number of passengers allowed), it closes and no longer takes on more passengers.

**Table 3.** Flights Departure Schedule.

| Flight No. | Flight | Departure Time | Gate |
|:---:|:---:|:---:|:---:|
| 1 | Flight A | 03:00 | 1 |
| 2 | Flight A | 05:00 | 2 |
| 3 | Flight C | 07:00 | 3 |
| 4 | Flight D | 08:00 | 2 |
| 5 | Flight E | 10:00 | 1 |
| 6 | Flight F | 11:00 | 3 |
| 7 | Flight G | 12:00 | 1 |
| 8 | Flight H | 13:00 | 3 |
| 9 | Flight I | 14:00 | 2 |
| 10 | Flight J | 16:00 | 1 |
| 11 | Flight K | 18:00 | 2 |
| 12 | Flight L | 20:00 | 3 |
| 13 | Flight M | 22:00 | 2 |

The following passenger will board the next available flight. It is important to emphasize that all passengers in the simulation must board their flights. In the model, as shown in Table 4, the total number of passengers during the simulation is also given as input. Therefore, setting the maximum number of passengers per flight also results in the number of flights. In the simulation presented in this research, the number of flights is 13 and is given in Table 3. The departure time and gate of each flight were randomly selected. The simulation starts two hours before the departure of the first flight.

**Table 4.** LGKM terminal simulation model input data.

| Input Data | Value |
|:---:|:---:|
| Number of passengers | 150 |
| Inter-arrival time for each entrance | 15 passengers per hour |
| Time spent at each store | 5 min |
| Time spent at WC | 2 min |

For the simulation, a list of passengers must be given by the user. Specifically, each passenger entering the model has the following characteristics:

- Carrier of COVID-19 (True or False);
- Online Check-in (True or False);
- Flight Number;
- Ticket Status (Business or Economy);
- Preferences (Other, Restaurant, Coffee Shop, Bathroom).

As in the simulation model of the transmission of COVID-19 among passengers in an airport area with stores, simulations were also carried out to compare the transmission of the virus for passengers visiting the shops inside the airport with and without routing. The simulations were carried out with percentages of infected passengers of 2%, 4%, 7%, 10%, 15%, and 20%.

*Results*

The airport simulation results are shown in Figures 7 and 8. And in this case study, it is observed that by routing the passengers to airport shops, for each percentage of infected passengers, there is a significant reduction in the transmission of COVID-19.

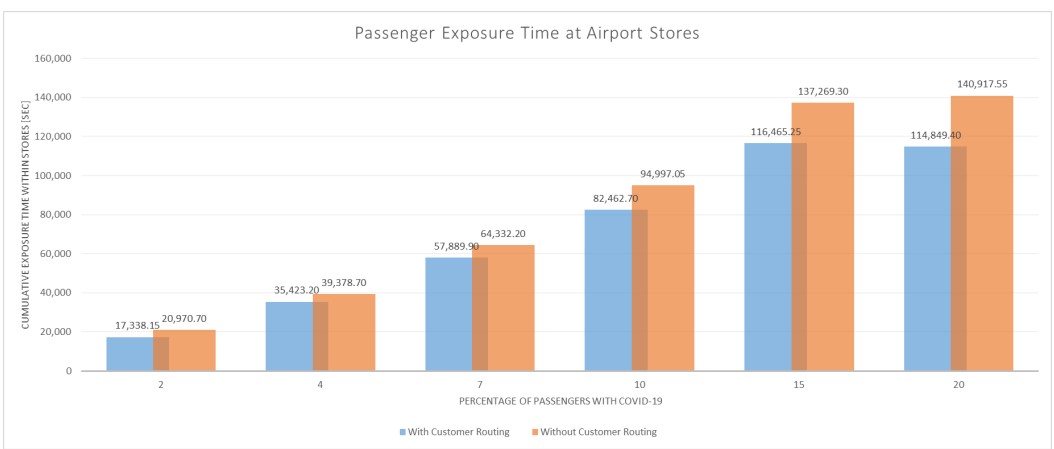

**Figure 7.** Passenger exposure time at airport.

| Infection Rate (%) | Without Passenger Routing | | | | | | | | With Passenger Routing | | | | | | | |
|---|---|---|---|---|---|---|---|---|---|---|---|---|---|---|---|---|
| | Other 1 | Other 2 | Restaurant 1 | Restaurant 2 | Coffee Shop 1 | Coffee Shop 2 | Bathroom 1 | Bathroom 2 | Other 1 | Other 2 | Restaurant 1 | Restaurant 2 | Coffee Shop 1 | Coffee Shop 2 | Bathroom 1 | Bathroom 2 |
| 2 | 18.75 | 20.45 | 20.70 | 20.20 | 11.40 | 11.05 | 9.90 | 8.90 | 15.80 | 13.55 | 16.95 | 17.35 | 9.10 | 8.00 | 7.80 | 7.05 |
| 4 | 20.20 | 17.10 | 20.45 | 19.00 | 10.35 | 10.15 | 9.25 | 9.10 | 16.05 | 13.90 | 16.55 | 17.05 | 9.15 | 7.75 | 7.80 | 7.30 |
| 7 | 18.70 | 19.60 | 21.50 | 20.70 | 11.70 | 11.10 | 9.40 | 9.85 | 15.80 | 13.40 | 17.95 | 17.80 | 9.40 | 7.75 | 7.95 | 7.25 |
| 10 | 17.40 | 19.75 | 23.40 | 21.65 | 10.05 | 9.95 | 9.60 | 10.00 | 15.60 | 13.95 | 17.80 | 18.00 | 8.60 | 7.60 | 8.05 | 7.20 |
| 15 | 18.65 | 20.30 | 22.55 | 22.75 | 11.05 | 11.15 | 9.90 | 0.00 | 15.85 | 13.85 | 18.35 | 18.50 | 9.10 | 8.25 | 8.00 | 7.15 |
| 20 | 18.50 | 20.40 | 20.50 | 22.60 | 10.95 | 10.35 | 10.25 | 9.40 | 16.05 | 14.30 | 16.65 | 16.70 | 8.45 | 7.90 | 7.75 | 7.05 |

**Figure 8.** Maximum number of passengers per store with and without routing.

## 6. Simulation Modeling of Transmission of COVID-19 at LGKM Kavala Airport

For the second simulation of the transmission of COVID-19 between passengers inside airport stores, we model the terminal of the LGKM Airport at Amygdaleonas Kavala. Figure 9 below shows the modeled floor plan of the terminal. This airport is used for training new pilots. Therefore, the terminal includes areas used for pilot briefings before their flight. In the context of the simulation, these spaces were modeled as airport terminal stores. Specifically, the terminal includes six different rooms and a bathroom. In the context of the simulation, the terminal was divided into three types of stores and two stores for

each type. Also, the model in this case simulates the option that a passenger wishes to go to the bathroom.

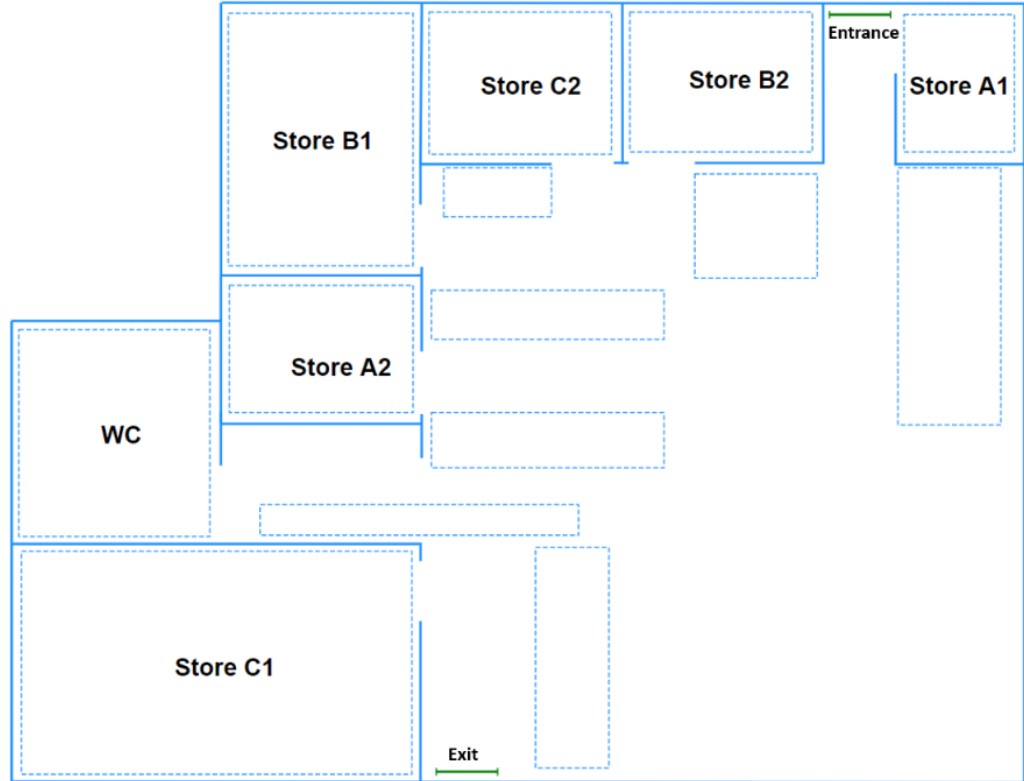

**Figure 9.** LGKM airport terminal floor plan.

The maximum capacity for each store was determined based on square meters (4 square meters per passenger) and each passenger can visit only one store of each specific type and only one time. The simulation model input data are displayed in Table 4.

*Results*

For the simulation, a sample of passengers was given as input to the model from which:

- 70% of passengers will want to visit Store A;
- 70% of passengers will want to visit Store B;
- 70% of passengers will want to visit Store C;
- 90% of passengers will want to visit the bathroom.

Regarding the percentage of passengers who have COVID-19, simulations were performed with percentages of passengers with infection rates of 2%, 4%, 7%, 10%, 15%, and 20%. Also, for each percentage of infected passengers, 40 simulations (20 simulations without passenger routing and 20 simulations with passenger routing) were performed and the results presented below are the averages of the results. These results are displayed in Figure 10. As can be seen from the results, the exposure time of each passenger in the stores was reduced by routing the passengers through the airport shops.

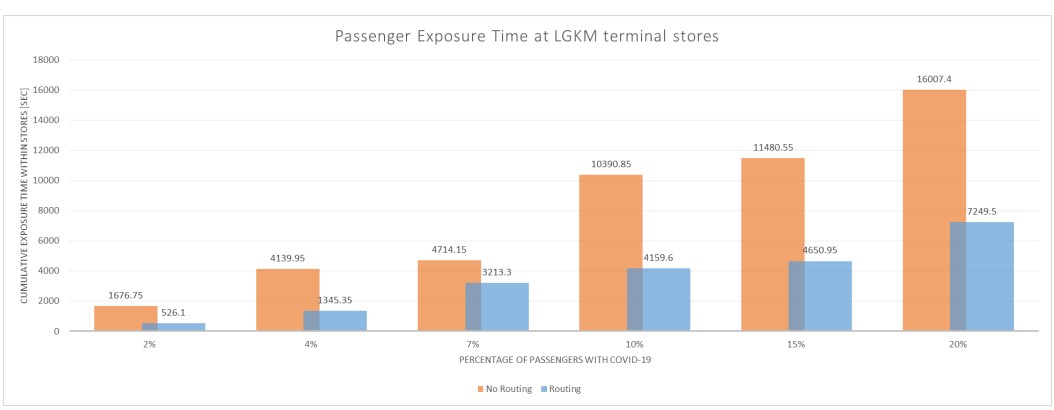

**Figure 10.** Passenger exposure time at the LGKM terminal stores.

## 7. Conclusions

In this paper, our use of the AnyLogic software enabled us to intricately simulate diverse scenarios within both an indoor complex and the intricate environment of an airport. The crux of our methodology lay in the incorporation of a straightforward yet potent weighting factor through our application, a factor that proved instrumental in achieving favorable outcomes by efficiently guiding passengers through various spaces.

Our conjecture revolves around the notion that the data required for this simulation were sourced from a combination of surveillance cameras and the preferences discerned from mobile phones. By synergizing these data streams, we constructed a comprehensive understanding of passenger behavior and flow within the simulated spaces.

This innovative approach strategically directed individuals towards communal areas within the airport, employing the weighted factor to optimize passenger routing. The paramount objective was to curtail potential virus exposure by minimising the time individuals spend in high-traffic zones. Through the seamless integration of technology, behavioral insights, and simulation prowess, our methodology offers a robust framework for enhancing the efficiency and safety of passenger movement in complex environments.

**Author Contributions:** Conceptualization, E.D.S. and D.T.; methodology, D.T.; software, D.T.; validation, E.D.S.; formal analysis, D.T.; investigation, E.D.S.; resources, V.K.; data curation, D.T.; writing—original draft preparation, D.T.; writing—review and editing, E.D.S. and D.T; visualization, D.T.; supervision, V.K.; project administration, E.D.S. and V.K; funding acquisition, V.K. All authors have read and agreed to the published version of the manuscript.

**Funding:** This research was done as part of the HAIKU project. This project has received funding from the European Union's Horizon Europe research and innovation programme HORIZON-CL5-2021-D6-01-13 under Grant Agreement no 101075332 but this document does not necessarily reflect the views of the European Commission.

**Data Availability Statement:** The raw data supporting the conclusions of this article will be made available by the authors on request.

**Conflicts of Interest:** The authors declare no conflict of interest.

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
