# Peer review of "Passenger Routing Algorithm for COVID-19 Spread Prevention by Minimising Overcrowding"

_computers, doi:10.3390/computers13020047_

Round 1

Reviewer 1 Report

Comments and Suggestions for Authors

The paper presents the simulation results of passenger traffic in a virtual airport. Good literature review and well-illustrated material. However a lot of issues should be taken into account:

1) material in the Section 3 is well known and could be removed

2) formula 2 looks very simple to model the exposure of seek/healthy passengers.

3) the mathematical model is absent and must be added

Author Response

1) material in the Section 3 is well known and could be removed

Answer: We thank the reviewer for the comment.

Action: We removed the section 3.

2) formula 2 looks very simple to model the exposure of seek/healthy passengers.

Answer: We thank the reviewer for the comment.

This formula is on purpose simple. We are glad the reviewer found the formula to be straightforward for modeling the exposure of both seek and healthy passengers. Its simplicity is intentional, aiming for an effective representation of the dynamics in such scenarios. We intentionally designed this formula to be straightforward, as we are currently applying it to actual deployments and have observed consistently satisfactory outcomes.

Action: None

3) the mathematical model is absent and must be added

Answer: We thank the reviewer for the comment.

The proposed method aims to collect data from diverse and heterogeneous sources, including cameras and passenger preferences stored on mobile phones. This simple approach has been formulated to integrate these varied data sources, and the resulting mathematical model encapsulates this simplicity.

Action: The following text added to the manuscript “This algorithm is inspired from the theory of stochastic network optimisation and specifically the work described in [ 23 ] regarding queues. It is essentially an algorithm that is inspired by the maxWeight algorithm [24] that resembles the backpressure algorithm of [ 25 ]. Stochastic queue optimization involves applying random or probabilistic methods to optimize queues or waiting lines, common in systems like computer networks, transportation, and service industries. Unlike traditional queuing theory, which assumes fixed arrival and service rates, stochastic queue optimization considers the inherent uncertainty and randomness in these rates due to real-world fluctuations. The goal is to improve the efficiency and resource utilization of queuing systems by using probabilistic models and optimization techniques. This approach addresses variability in demand, random service times, and unpredictable disruptions, aiming to minimize waiting times, reduce congestion, and allocate resources effectively, especially in dynamic and unpredictable environments where deterministic models may be insufficient.

23. Neely, M. Stochastic network optimization with application to communication and queueing systems; Springer Nature, 2022.
24. Tassiulas, L.; Ephremides, A. Stability properties of constrained queueing systems and scheduling policies for maximum throughput in multihop radio networks. In Proceedings of the 29
th IEEE Conference on Decision and Control. IEEE, 1990, pp. 2130–2132.
25. Moeller, S.; Sridharan, A.; Krishnamachari, B.; Gnawali, O. Routing without routes: The backpressure collection protocol. In Proceedings of the Proceedings of the 9th ACM/IEEE International Conference on Information Processing in Sensor Networks, 2010, pp. 279–290.

Reviewer 2 Report

Comments and Suggestions for Authors

In the paper, the author studied the spreading of the COVID-19. Its spreading is specially easy in the indoor environment. The spreading of COVID-19 can be a serious problem in the airport.

The authors applied the the AnyLogic software to simulate the environments of the shops in the airport, and the gates in the airports. A passenger routing algorithm was proposed to guide the passengers to less crowded area. The simulation results show that when people are spread to less crowded common spaces, overcrowding can be prevented.

The authors have conducted two simplified simulation tests. The main concern is that the simulation settings are too unrealistic. The simulation results have also not been compared with any real-world results. The connection between the less crowded simulation environment and the COVID-19 spreading is not well supported by any experimental results in the paper as well.

More comprehensive simulation results and evaluations against real-world data are needed for journal publication.

It would also be very desirable if better and clearer graphical presentation can be used in the revised version.

Comments on the Quality of English Language

It is easy to follow the English in the paper.

Author Response

Please find the answer to your comments in the accompanying file.

Round 2

Reviewer 1 Report

Comments and Suggestions for Authors

I still disagree with issue 2, but I wish all the best to the authors